# Neo-Adjuvant Therapy for Metastatic Melanoma

**DOI:** 10.3390/cancers16071247

**Published:** 2024-03-22

**Authors:** Anke M. J. Kuijpers, Alexander C. J. van Akkooi

**Affiliations:** 1Department of Surgical Oncology, Netherlands Cancer Institute, 1066 CX Amsterdam, The Netherlands; a.kuijpers@nki.nl; 2Melanoma Institute Australia, Sydney, NSW 2060, Australia; 3Faculty of Medicine and Health, University of Sydney, Sydney, NSW 2050, Australia; 4Department of Melanoma and Surgical Oncology, Royal Prince Alfred Hospital, Sydney, NSW 2050, Australia

**Keywords:** melanoma, neo-adjuvant, immunotherapy

## Abstract

**Simple Summary:**

Neo-adjuvant systemic (NAS) therapy for melanoma is leading the way in immunotherapy for oncology. There is both a strong biological rationale to support NAS therapy, as well as exciting results in terms of superior efficacy to the current standards of care. Simply giving pembrolizumab 3 doses prior to surgery and 14 after rather than all 17 courses as adjuvant therapy only, demonstrated a 23% improvement in event-free survival (EFS). Combination immunotherapy with ipilimumab and nivolumab seems even more potent with ~60% of patients achieving Major Pathologic Responses (MPR) vs. ~25–30% for single agent anti-PD-1. NAS therapy will allow for a tailored approach to the therapy of melanoma patients, which includes a potential safe de-escalation of the extent of surgery and discontinuation of systemic therapy for MPR patients. And early identification of non-responders who can be directed towards clinical trials with novel therapies. These developments are now splashing over to other solid tumors, but there is still a need to develop a bespoke panel of biomarkers to further improve upon the current treatments. NAS therapy will soon become the novel standard of care approach for macroscopic stage III melanoma.

**Abstract:**

Melanoma treatment is leading the neo-adjuvant systemic (NAS) therapy field. It is hypothesized that having the entire tumor in situ, with all of the heterogeneous tumor antigens, allows the patient’s immune system to have a broader response to the tumor in all its shapes and forms. This translates into a higher clinical efficacy. Another benefit of NAS therapy potentially includes identifying patients who have a favorable response, which could offer an opportunity for the de-escalation of the extent of surgery and the need for adjuvant radiotherapy and/or adjuvant systemic therapy, as well as tailoring the follow-up in terms of the frequency of visits and cross-sectional imaging. In this paper, we will review the rationale for NAS therapy in resectable metastatic melanoma and the results obtained so far, both for immunotherapy and for BRAF/MEKi therapy, and discuss the response assessment and interpretation, toxicity and surgical considerations. All the trials that have been reported up to now have been investigator-initiated phase I/II trials with either single-agent anti-PD-1, combination anti-CTLA-4 and anti-PD-1 or BRAF/MEK inhibition. The results have been good but are especially encouraging for immunotherapies, showing high durable recurrence-free survival rates. Combination immunotherapy seems superior, with a higher rate of pathologic responses, particularly in patients with a major pathologic response (MPR = pathologic complete response [pCR] + near-pCR [max 10% viable tumor cells]) of 60% vs. 25–30%. The SWOG S1801 trial has recently shown a 23% improvement in event-free survival (EFS) after 2 years for pembrolizumab when giving 3 doses as NAS therapy and 15 as adjuvant versus 18 as adjuvant only. The community is keen to see the first results (expected in 2024) of the phase 3 NADINA trial (NCT04949113), which randomized patients between surgery + adjuvant anti-PD-1 and two NAS therapy courses of a combination of ipilimumab + nivolumab, followed by surgery and a response-driven adjuvant regimen or follow-up. We are on the eve of neo-adjuvant systemic (NAS) therapy, particularly immunotherapy, becoming the novel standard of care for macroscopic stage III melanoma.

## 1. Introduction

Neo-adjuvant systemic (NAS) therapy has been around for many decades for different types of solid tumors [1]. Obviously, these first experiences are related to NAS therapy in use with more classical therapies, such as chemotherapy, and not with modern therapies, such as immune checkpoint inhibitors (ICIs).

Importantly, NAS therapy is, per definition, related to cases who will proceed to undergo surgery at a later timepoint. Attempting to convert unresectable disease into resectable disease is referred to as ‘induction’ therapy. This means that the extent of surgery can be tailored to the response to NAS therapy. Drawing a parallel with breast cancer, a good response to NAS therapy might render a mastectomy no longer required and offer the possibility of a lumpectomy instead [2].

Melanoma has been key in the immunotherapy development field, with the discovery of anti-CTLA-4 (ipilimumab) being effective in treating stage III unresectable and stage IV metastatic melanoma (also known together as ‘advanced melanoma’) [3]. Before this discovery, the prognosis of advanced melanoma patients was poor and comparable to that of stage IV non-small-cell lung cancer (NSCLC) or pancreatic cancer [4].

Another crucial discovery in advanced melanoma has been the fact that approximately 45% of patients harbor a BRAF mutation within their tumor cells. This driver mutation in the MAPK pathway can be effectively targeted by using selective BRAF/MEK inhibitors [5,6].

In this paper, we will review the rationale for NAS therapy in resectable metastatic melanoma and the results obtained so far, both for immunotherapy and for BRAF/MEKi therapy, and discuss the response assessment and interpretation, toxicity and surgical considerations.

## 2. Rationale for Neo-Adjuvant as Compared to Adjuvant Systemic Therapy

Melanoma is leading the NAS immunotherapy field with clinical studies and other solid tumors are more recently following this example. However, the pre-clinical models developed by Teng et al. tested NAS immunotherapy first in breast and colorectal cancer in mice and compared it to adjuvant immunotherapy [7]. They found a higher efficacy for NAS immunotherapy. It is hypothesized that having the entire tumor in situ, with all of the heterogeneous tumor antigens, allows the patient’s immune system to have a broader response to the tumor in all its shapes and forms. This translates into a higher clinical efficacy. Figure 1 illustrates NAS vs. adjuvant therapy.

Besides this potential advantage, the International Neo-adjuvant Melanoma Consortium (INMC) has postulated a few proposed advantages of the NAS therapy model. These include identifying patients who have a favorable response, which could offer an opportunity for the de-escalation of both the extent of surgery and the need for adjuvant radiotherapy and/or adjuvant systemic therapy, as well as tailoring the follow-up in terms of the frequency of visits and cross-sectional imaging [8]. The response and/or toxicity associated with NAS therapy might give important prognostic and predictive information that might be useful in the choice of adjuvant systemic therapy. Some other proposed benefits of NAS therapy are that it allows patients with resistant disease to be recognized early and channeled toward clinical trials; it can be an excellent model for quick drug development; NAS therapy can be used as a platform for biomarker discovery, and, last but not least, there are no delays in initiating effective systemic therapy [8].

NAS therapy does not only have advantages: some have also pointed out potential critical downsides, which need to be considered. First and foremost, patients with a poor response could lose their window of opportunity for a complete curative resection to be performed with clear margins. The drug-related toxicity might be higher than in adjuvant patients, and this could impact long-term morbidity and QoL. Drug-related toxicity could require steroids or other treatments that can potentially delay surgical resection and/or increase the risk of undergoing surgery.

## 3. Results Obtained to Date

### 3.1. Neo-Adjuvant Immunotherapy in Stage III and Stage IV Resectable Melanoma

The evidence for the benefit of NAS is growing, although no phase III trials have been published yet. Several regimens for NAS are currently being investigated, for instance, regimens with single-agent anti-PD1 therapy, combination therapy with anti-PD1 and anti-CTLA4 and other combination therapies. Table 1 summarizes the NAS ICI studies. Recently, a practice-changing phase II trial (SWOG 1801) has been published that showed a significantly longer event-free survival (EFS) with the neo-adjuvant administration of pembrolizumab combined with adjuvant administration compared to the adjuvant alone. It is important to realize that EFS is a more appropriate endpoint to assess when comparing neo-adjuvant and adjuvant strategies, because it covers all randomized patients, including those who progressed prior to surgery, whereas relapse-free survival (RFS) only reports on those who underwent surgery. Patients randomized to the neo-adjuvant group received an intravenous infusion of 200 mg of pembrolizumab every 3 weeks for a total of three doses before surgery, followed by an additional 15 doses of pembrolizumab as adjuvant therapy after surgery. In the patients who were randomized to the adjuvant-only group, surgery was followed by the adjuvant intravenous infusion of 200 mg of pembrolizumab every 3 weeks for 18 doses. Patel et al. showed that EFS at 2 years was 72% in the neo-adjuvant group, compared to 49% in the adjuvant-only group [9]. A pathologic complete response (pCR) was seen in 38%. Table 1 summarizes the results from NAS ICI therapy trials in melanoma to date.

Previous studies with neo-adjuvant anti-PD1 showed similar response rates of around 25–30% [10,11]. Few long-term results are available; however, the numbers that are available are very promising for patients with a complete pathologic response (pCR) or near-complete pathological response (near-PCR; max. 10% viable tumor cells), which are also lumped together as a major pathologic response (MPR). A recent update by Sharon demonstrated, with a median follow-up of 61.9 months, a 5-year OS of 100% for neo-adjuvant responders and an RFS of 75% after only a single dose of 200 mg pembrolizumab [12].

Several trials have been performed with the goal of finding the best neo-adjuvant strategy with the combination of anti-PD1 and anti-CTL4. In the phase 1b Optimal Neo-Adjuvant Combination Schema of Ipilimumab and Nivolumab (OpACIN) study, patients were randomized to neo-adjuvant or adjuvant therapy to analyze the ability of expanding tumor-specific T-cell clones [13]. Patients were randomized to upfront surgery followed by four combination doses of ipilimumab 3 mg/kg and nivolumab 1 mg/kg (IPI/NIVO) every 3 weeks or a sandwich construction with two doses of the combination immunotherapy followed by surgery and then by two additional doses of the combination. It was concluded that administering the combination immunotherapy before surgery was feasible for all patients. Significantly more tumor-specific T-cell clones were found in the circulation of the neo-adjuvant patients compared to the adjuvant patients. A pathologic response rate of 78% was achieved in the nine patients who were treated with neo-adjuvant immunotherapy. However, grade 3 or 4 toxicity events were highly prevalent in both groups (90%).

Thus, a subsequent trial was designed to reduce toxicity and keep the same efficacy by finding the optimal dosing schedule of the combination. The subsequent trial was the OpACIN-neo trial, a phase II study assessing different combination therapy regimens for neo-adjuvant ipilimumab and nivolumab [14]. In this study, patients were randomized to different neo-adjuvant IPI/NIVO strategies, followed by surgery. The dosing schedules contained IPI 3 mg/kg and NIVO 1 mg/kg for arm A and IPI 1 mg/kg and NIVO 3 mg/kg for arm B. In arm C, patients received sequential IPI 3 mg/kg and NIVO 3 mg/kg. The Overall Response Rates (ORRs) in this study were 80%, 77% and 65%, respectively, with a pCR rate of 50–60%. The arm with the fewest grade 3–4 AEs was arm B with the lower ipilimumab dose (20%).

An extension cohort was added to the OpACIN-neo trial to confirm the efficacy of OpACIN-neo arm B, as well as address the question of treatment de-escalation. In this trial, the PRADO trial, a major pathological response rate of 61% and a 2-year relapse-free survival of 93% were reported in patients with a major pathological response [15]. A recent survival update of the OpACIN and OpACIN-neo trials combined showed that after a median follow-up of 69 months for the OpACIN study, 1/7 patients with a pathologic response had disease recurrence [16]. In the OpACIN-neo study, after a median follow-up of 47 months, the estimated 3-year RFS and OS rates were 82% and 92%, respectively. For the patients in this study who had a pathologic response, the estimated 3-year RFS was 95%. The pathologic response was the strongest predictor of recurrence in a multiple regression analysis of the total patients treated with NAS in both trials combined.

Currently, the NADINA trial (NCT04949113) is being carried out. In this phase III international multi-center trial, patients are being randomized to standard single-agent anti-PD-1 adjuvant therapy (control arm) or neo-adjuvant combination immunotherapy. The neo-adjuvant patients receive two cycles of neo-adjuvant nivolumab 240 mg and ipilimumab 80 mg every 3 weeks with a therapeutic lymph node (TLND) dissection in week 6. Post-surgery adjuvant therapy is only administered to patients with a pathologic partial (pPR) or non-response (pNR). Adjuvant therapy consists of 11 cycles of nivolumab 480 mg every 4 weeks for BRAF-wildtype (wt) patients with a pPR or pNR. BRAF V600E/K mutation-positive pPR or pNR patients receive adjuvant dabrafenib 150 mg bid and trametinib 2 mg qd for 46 weeks. Patients randomized to the adjuvant arm receive 12 courses of nivolumab 480 mg every 4 weeks post-surgery. The first results are expected in 2024.

Two trials compared neo-adjuvant combination therapy to single-agent neo-adjuvant anti-PD1. The first study, single-center phase 2, compared neo-adjuvant nivolumab 3 mg/kg monotherapy every 2 weeks up to four doses to neo-adjuvant combination immunotherapy with ipi 3 mg/kg and nivo 1 mg/kg up to three doses. Both arms received adjuvant nivolumab post-surgery for 12 months. The ORR of the single-agent NIVO group was 25%, with all of these patients having a pCR. In the IPI/NIVO group, the ORR was 73%, with a 45% pCR rate. After a median follow-up of 15 vs. 15.6 months, patients with IPI/NIVO had better outcomes than the single-agent NIVO patients; however, the differences in OS, PFS and DMFS did not reach statistical significance. This study was stopped prematurely, because 2/12 patients in the NIVO group progressed both in the regional field and systemically, but high rates of grade 3 toxicity progressed in the IPI/NIVO group too.

Relatlimab is another fairly new immunotherapeutic agent targeting LAG-3. The combination of anti-PD1 and anti-LAG-3 showed promising results in the treatment of advanced melanoma [17]. Amaria et al. integrated relatlimab combined with nivolumab into a neo-adjuvant regimen [18]. In this study, patients received two neo-adjuvant doses of nivolumab 480 mg and relatlimab 160 mg intravenously every 4 weeks, followed by surgery. Postoperatively, patients received another ten doses of this combination therapy. The pathologic complete response rate was 57%, with a 70% ORR, resulting in 1- and 2-year recurrence-free survival rates of 100% and 92%, respectively, for patients with any pathologic response. This regimen was considered safe, with no grade 3–4 toxicity from the neo-adjuvant combination therapy.

### 3.2. Neo-Adjuvant Targeted BRAF/MEK Inhibitors for BRAF-Mutated Melanoma in Stage III and Stage IV Resectable Melanoma

Two studies were performed on BRAF/MEK inhibitors in the neo-adjuvant setting. The first study was a single-center randomized phase 2 study that randomized patients 1:2 to either surgery followed by adjuvant therapy (up to the treating physician) or to 8 weeks of neo-adjuvant dabrafenib 150 mg bid and trametinib 2 mg per day (DAB/TRAM), followed by surgery and postoperative DAB/TRAM for up to 44 weeks [19]. The study by Amaria et al. was closed prematurely, as the interim safety analysis showed an excess event rate in the standard arm. Therefore, only 7 patients were included in the standard arm and 14 patients in the DAB/TRAM intervention arm. After a median follow-up period of 18.6 months, 10 of 14 patients with neo-adjuvant therapy and 0 of 7 patients in the standard arm were alive without evidence of disease. The median event-free survival period was 19.7 months vs. 2.9 months, respectively (HR 0.016, 95% CI 1.7–not estimable) [19]. It must be noted that only one patient in the standard arm received adjuvant therapy. This study showed that neo-adjuvant plus adjuvant DAB/TRAM significantly improved event-free survival compared to the standard of care (basically surgery alone) in clinical stage 3 and 4 resectable melanoma. The study continued as a single-arm trial, and further results are expected in the near future.

The second study reported the outcome of the NeoCombi trial, which was a single-arm, open-label, single-center, phase 2 study that assessed neo-adjuvant DAB/TRAM (common dose) for 12 weeks and postoperative DAB/TRAM for 40 weeks [20]. In total, 35 patients were enrolled, of whom none were progressive during neo-adjuvant treatment. All of them showed a pathological response: 49% had a complete pathological response. Respectively, the 1- and 2-year relapse-free survival rates were 82.4% and 63.3% for patients with a complete pathological response. Of the patients with a non-complete pathological response, the 1- and 2-year relapse-free survival rates were 72.2% and 24.4% [20].

### 3.3. Neo-Adjuvant Combination Targeted and Immunotherapy

Recently, the NeoACTIVATE phase II trial reported on patients who were randomized to receive either 12 weeks of neo-adjuvant vemurafenib, cobimetinib and atezolizumab (BRAF-mutated, Cohort A, *n* = 15) or cobimetinib and atezolizumab (BRAF-wildtype, Cohort B, *n* = 15), followed by TLND and 24 weeks of adjuvant atezolizumab [21]. The trial so far has reported the response rates, showing an MPR rate of 66.7% for BRAF mutant and 33.3% for BRAF-wildtype patients. Grade ≥ 3 AEs occurred in 63% of patients during neo-adjuvant treatment [21]. It will be important to see whether the higher MPR rate found for this combination of targeted and immunotherapy will show durable efficacy in terms of EFS or RFS. Triplet therapy has not shown a clinically meaningful benefit for the treatment of advanced melanoma [22], but this could be different for neo-adjuvant therapy in stage III melanoma.

## 4. Response Assessment and Interpretation

### 4.1. Response Assessment during Treatment

To identify patients who do not respond or even progress during treatment, it is important to perform a response evaluation during or after treatment. This can be conducted by either radiographic or pathologic assessment. Several studies have reported on the discordance between radiographic and pathologic assessments of response. Generally, response is underestimated by cross-sectional imaging modalities [10,13,14,23]. Generally, contrast-enhanced CT is recommended for response assessments during neo-adjuvant systemic treatment [24]. Functional imaging techniques based on new immune-targeted epitopes are currently being investigated [25].

Another method of response assessment is biomarker analysis. There are a few biomarkers that have predictive value for response in the neo-adjuvant setting, such as the IFN-gamma signature and tumor mutational burden; however, no definite biomarker or combination of biomarkers has yet been established [20,23].

### 4.2. Pathological Response Assessment

Since radiographic methods have not yet been able to predict response or, even more interesting, complete response, pathologic evaluation is still mandated. This is primarily performed by removing the complete lymph node basin with TLND or oligometastatic visceral tumor localization. However, the index lymph node (ILN) technique, described below, is also a novel method of pathologic response assessment that holds promise for the future. There has been an ongoing discussion in the pathologic field on how to assess response after neo-adjuvant therapy for melanoma. This is mainly due to the different patterns of histological response that can be identified in treated tissues. Also, the response to NAS can be heterogeneous within the same lymph node, complicating the drawing of firm conclusions based on a histological biopsy only (sampling error). Therefore, the INMC has developed a set of standardized criteria to analyze the pathologic response in surgical specimens [26,27]. This consensus is very much warranted, because of the prognostic value of the response rate, the ability to guide further therapy based on the response and the establishment of the response as a valid surrogate endpoint for neo-adjuvant trials.

## 5. Toxicity

An interesting observation regarding toxicity was made in the OpACIN study. This study demonstrated a rate of 90% grade 3/4 adverse events (AEs) with ipilimumab 3 mg/kg and nivolumab 1 mg/kg Q3W [13]. This toxicity meant that only 1/10 patients in each arm completed all four courses of IPI/NIVO, and the majority only received two or three courses. The Amaria et al. trial found a similarly high rate of 73% grade 3/4 AEs [10]. This same dose schedule for IPI/NIVO is used for advanced melanoma and, in the pivotal Checkmate 067 trial, was associated with a mere rate of 55% grade 3/4 AEs [28]. It has been hypothesized that the reason for this higher rate of toxicity that was observed might be due to a less inhibited host immune system, which requires less ICI to re-start an antitumor response, but it will also more easily lead to AEs.

Amaria et al. and Huang et al. reported 8% and 0% grade 3/4 AEs for single-agent anti-PD-1 with nivolumab and pembrolizumab, respectively [10,11]. Subsequently, the OpACIN-neo and PRADO trials used the ‘low-dose’ or ‘flip-dose’ of IPI/NIVO (ipilimumab 1 mg/kg and nivolumab 3 mg/kg) and found that the rates of grade 3/4 AEs were 20% and 22%, respectively [14,15]. The SWOG-1801 study reported grade 3/4 AE rates of 12% and 14% for the neo-adjuvant and adjuvant single-agent anti-PD-1 (pembrolizumab), respectively [9].

Finally, Amaria et al. reported no grade 3/4 AEs for the combination of nivolumab and relatlimab [18]. To summarize, early NAS trials with the full-dose IPI/NIVO combination encountered high rates of severe AEs, but more recent trials with either single-agent anti-PD-1, the ‘low-dose’ or ‘flip-dose’ of IPI/NIVO and/or the combination with anti-LAG-3 have presented more feasible rates of grade 3/4 AEs.

## 6. Surgical Considerations

Several anecdotal cases of surgery after immunotherapy have raised concerns within the surgical community about tissue becoming more fibrous and, therefore, making it more difficult to find the anatomical plane to perform a resection, thereby causing more unnecessary damage to nearby structures such as arteries, veins, nerves and other organs. This could be reflected in a higher rate of surgical complications after immunotherapy, more blood loss and/or a longer duration of surgery. It seems that these anecdotal cases are more frequent after a case that was initially assessed to be an advanced melanoma case becomes resectable after a longer duration of immunotherapy rather than a short defined NAS period.

Unfortunately, most NAS therapy trials have not captured these specific surgical endpoints. A recent analysis by Zijlker et al. investigated this in NAS patients who were treated in Amsterdam within the OpACIN and OpACIN-neo studies and compared those patients to a cohort of patients from Amsterdam who underwent upfront surgery. They found no statistically significant differences between the groups in terms of overall complications (31.8% vs. 36.8%, *p* = 0.578) or for specific morbidities such as seroma (56.8% vs. 57.9%, *p* = 0.908) or lymphedema (22.7% vs. 13.2%, *p* = 0.175) [29]. The surgery was slightly longer after NAS therapy, albeit, again, not statistically significant (105 vs. 90 min, *p* = 0.077). Zijlker et al. also looked at textbook outcomes, which is a contemporary composite measure of surgical outcomes that includes no grade II-V complications, R0 resection, no prolonged hospital stay, no re-take to theater and no re-admission. They found no differences in textbook outcomes (50% vs. 49%, *p* = 0.889) [29].

A study by Hieken et al. evaluated the surgical experience in cases after NAS therapy. The results were obtained from the NeoACTIVATE trial, which offered patients either vemurafenib, cobimetinib and atezolizumab (vem/cobi/atezo) or cobimetinib and atezolizumab (cobi/atezo). They found that the degree of difficulty increased in 17% but decreased in 25% of cases [30]. However, patients did not exclusively receive ICI: they all also received targeted therapy.

One of the hot topics after effective NAS therapy for all solid tumors is the ability to de-escalate surgery and/or be organ-sparing for patients who achieve an excellent response. As a side study to the OpACIN-neo study, Schermers et al. reported on the technique to selectively identify and remove the index lymph node (ILN) and assess the response within the ILN to the rest of the nodal basin. The ILN was defined as the largest lymph node metastasis at baseline. A magnetic marker was placed into the ILN before starting any NAS therapy in order for the surgeon to be able to identify the correct ILN during surgery. At surgery, the ILN was selectively removed, followed by the scheduled therapeutic lymph node dissection (TLND). Schermers et al. demonstrated a 100% congruency between the ILN response and the rest of the TLND specimen in this small prospective pilot study. Subsequently, Reijers et al. retrospectively analyzed the ILN within the TLND specimen and confirmed the high accuracy of ILN in representing the response to NAS therapy in 81/82 patients [31].

This concept of the ILN was used prospectively within the PRADO study, where patients did not undergo TLND but only ILN resection to assess the response to NAS therapy. Patients who did not achieve an MPR underwent TLND in a second session. Only 4/60 MPR patients recurred, and 3/4 recurrences were in the nodal basin, which could be dealt with by performing delayed TLND [15]. More importantly, surgical morbidity was much reduced for the ILN patients compared to TLND patients (46% vs. 84% complications, *p* < 0.001), and this was especially true for lymph edema (5% vs. 39%, *p* < 0.001) [15]. This translated into a clearly improved health-related quality of life (HRQoL) for ILN patients in general and for specific functioning domains too [15]. However, despite excitement within the international community, PRADO is seen as a proof-of-concept trial, which was too small and immature to definitively change practice yet, as reflected by the INMC statement and guidelines [8,32]. Efforts are ongoing to set up a suitable trial that will establish the ILN approach as the novel standard of care for MPR patients after NAS immunotherapy in melanoma.

The INMC recommends using surgical questionnaires (baseline and after 24 h), which were established through consensus, to assess the ease of surgery after NAS in any NAS therapy trials that are being conducted [8]. The NADINA trial is using these questionnaires prospectively.

## 7. Discussion

Neo-adjuvant systemic therapy is a promising treatment strategy for high-risk stage III and resectable stage IV melanoma patients. Several phase I–II studies have been performed on different neo-adjuvant immunotherapeutic and targeted treatment regimens [9,10,11,13,14,18,19,20]. Most of these studies focused on the efficacy and safety of NAS considering systemic toxicity and the effects on subsequent surgery. No detrimental effects on the quality of the surgery, operating time or postoperative complications were found [29]. In most studies, only a few patients (~7–8%) progressed during NAS to the extent that operating was no longer feasible or beneficial, most with the detection of distant visceral disease. This could be regarded as losing the window of opportunity for surgical resection; however, it most likely reflects a more aggressive tumor biology, and thus, upfront surgery would have been futile too. Overall, patients with a pathological complete or near-complete response (MPR) have a very good prognosis; however, long-term data and/or phase III trials are lacking.

The main question is whether NAS is superior to adjuvant therapy for patients with macroscopic resectable disease. Other questions include whether we need both NAS AND adjuvant systemic therapy, or whether neo-adjuvant therapy is sufficient in patients with an MPR. The SWOG 1801 trial showed superior EFS for patients with neo-adjuvant and adjuvant pembrolizumab compared to adjuvant pembrolizumab alone [9]. These results are already quite convincing, which has prompted the latest ASCO guidelines to recommend its use, but formal phase 3 evidence is pending [32]. Whether patients in this trial needed adjuvant therapy after a pCR or MPR, we cannot know, since all patients proceeded to receive adjuvant pembrolizumab.

The currently running NADINA trial (NCT04949113) was designed with a response-based adjuvant treatment strategy. This study randomizes between neo-adjuvant IPI/NIVO and adjuvant nivo only. Patients with an MPR do not receive further adjuvant treatment. For patients with a partial or no response, adjuvant treatment is based on BRAF mutation status. The first results of this study are to be expected in 2024. This study can help cement neo-adjuvant immunotherapy as the novel standard of care for macroscopic stage III melanoma. If successful, it will show that short-term neo-adjuvant therapy alone is effective, and there is no need for the completion of a full year of adjuvant therapy, which will be highly beneficial to reduce the usage of healthcare resources and thereby improve costs.

Which neo-adjuvant strategy is the best for which patient will be the next point for debate. IPI/NIVO has shown superior MPR rates to single-agent anti-PD-1 and seems to incur a higher rate of EFS. However, since the combination IPI/NIVO regimen is associated with a higher rate of grade 3/4 adverse events, patients who would have achieved a pCR with anti-PD1 alone are being overtreated. Unfortunately, an upfront response prediction of NAS is a challenge without well-established biomarkers. Although the gamma-interferon signature seems to be able to select patients who are more susceptible to anti-PD1 alone, this needs to be further validated [33].

For BRAF-mutant patients, the option of neo-adjuvant targeted therapy is available as well. The neo-adjuvant studies for BRAF/MEK inhibitors show very good response rates; however, EFS rates in patients with a pCR appear to be lower than in patients with a pCR after neo-adjuvant immunotherapy. The quality of a pCR is not the same for targeted therapy as for immunotherapy. Currently, combination regimens of BRAF/MEK inhibitors concurrent with or sequential to anti-PD1 versus anti-PD1 alone are being investigated in the NeoTrio trial [34]. The initial data, however, did not seem to indicate any improvement by adding targeted therapy to immunotherapy, similar to the disappointing results for triplet therapies for metastatic melanoma [35,36,37].

## 8. Future Perspectives

In some countries, NAS has been available for patients since the first promising results of the SWOG trial due to the fact that using this schedule is net neutral in terms of costs to payers but might induce a significant benefit. In other countries, the results of phase 3 trials are awaited. All in all, it is expected that, in the next couple of years, NAS will become common practice for (a subset of) resectable stage III and IV melanoma patients. This will also give us the opportunity to use the neo-adjuvant platform for further research into response monitoring and predictive biomarkers and to identify patients for whom we need to develop other treatment strategies.

One interesting recent discovery is that of pretreatment emotional distress (ED) as a biomarker. It is hypothesized that ED can negatively influence the patient’s immune system through β-adrenergic or glucocorticoid signaling [38,39]. From the PRADO study, a post hoc analysis demonstrated that pretreatment ED was significantly associated with reduced MPR rates (46% vs. 65%, HR 0.20, *p* = 0.038) [39]. This remained true after correcting for other known prognostic factors, such as the interferon-gamma signature and tumor mutational burden [39].

At the same time, the discoveries made in melanoma have also been applied to other cancer types, such as microsatellite instability (MSI), high colorectal cancer, cutaneous squamous cell carcinoma and urothelial cancer [40,41,42].

In addition, we need to put effort into selecting patients in whom it is possible to de-escalate surgery. Most NAS trials in melanoma were performed in patients who received highly morbid TLND, which has an enormous impact on the quality of life of these melanoma patients. For other cancer types, there is the opportunity to be organ-sparing, for example, in colorectal and bladder cancer, without the need to remove the colon, rectum or bladder, which has obvious HRQoL benefits to patients. Potentially avoiding surgery altogether might be possible in the future. With such promising results for patients achieving pCR after NAS, it is imperative to study the possibilities of de-escalation in the near future.

## 9. Conclusions

We are on the eve of neo-adjuvant systemic (NAS) therapy, particularly immunotherapy, becoming the novel standard of care for macroscopic stage III melanoma. Future endeavors should focus on the identification of biomarker(s) to personalize regimens, de-escalating surgery where possible and de-escalating adjuvant therapy where possible, and the development of novel drugs (combinations) for pathologic non-responders.

## Figures and Tables

**Figure 1 cancers-16-01247-f001:**
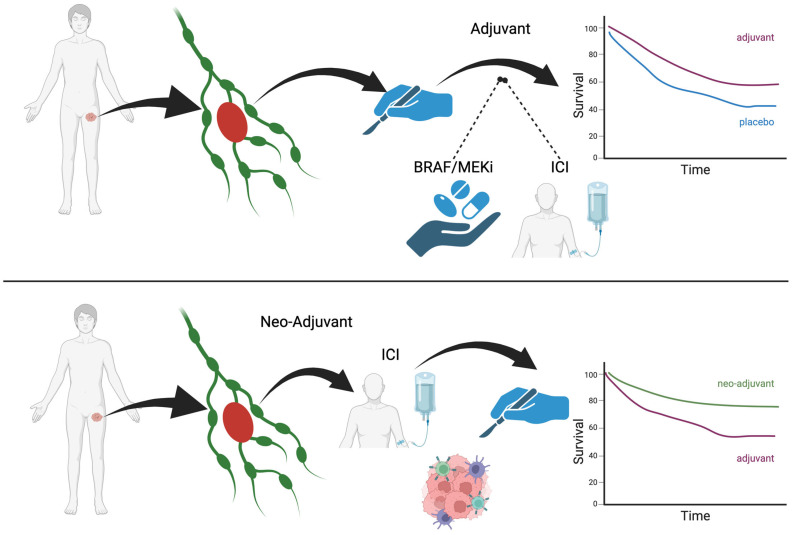
NAS vs. adjuvant therapy.

**Table 1 cancers-16-01247-t001:** Summary of neo-adjuvant immunotherapy trials in melanoma.

Study	Median FUP	*n*	Treatment Regimen(s)	pCR *n* (%)	MPR *n* (%)	ORR *n* (%)	TRAE ≥ Grade 3	RFS	EFS	DMFS	OS
OpACIN, Blank et al.	25.6 months	20	Adjuvant Arm: Max 4 × ipilimumab (3 mg/kg) + nivolumab (1 mg/kg) Q3W Neo-Adjuvant Arm: Max 2 × ipilimumab (3 mg/kg) + nivolumab (1 mg/kg) Q3W + max 2 more courses after surgery	N/A 3 (33%)	N/A 6 (67%)	N/A 7 (78%)	90% 90%	2 yr 60% 2 yr 80%	N/A	N/A	2 yr 80% 2 yr 90%
OpACIN, Versluis et al.	69 months	20	Adjuvant Arm: Max 4 × ipilimumab (3 mg/kg) + nivolumab (1 mg/kg) Q3W Neo-Adjuvant Arm: Max 2 × ipilimumab (3 mg/kg) + nivolumab (1 mg/kg) Q3W + max 2 more courses after surgery	N/A 3 (33%)	N/A 6 (67%)	N/A 7 (78%)	90% 90%	5 yr 60% 5 yr 70%	5 yr 60% 5 yr 70%	5 yr 60% 5 yr 80%	5 yr 70% 5 yr 90%
Huang et al.	25 months	30	Single-dose pembrolizumab 200 mg prior to surgery + max one year of adjuvant pembrolizumab 200 mg Q3W post-surgery	25%	~30%	N/A	<30%	2 yr 63%	N/A	N/A	2 yr 93%
Sharon et al.	61.9 months	30	Single-dose pembrolizumab 200 mg prior to surgery + max one year of adjuvant pembrolizumab 200 mg Q3W post-surgery	5 (16.7%)	8 (26.7%)	N/A	<30%	5 yr 75% MPR 63.6% non-MPR	N/A	N/A	5 yr 100% MPR 72.8% non-MPR
Amaria et al.	15.6 months	23	Arm A: max 4 × neo-adjuvant nivolumab monotherapy (3 mg/kg) Q2W Arm B: neo-adjuvant combination of max 3 courses of ipilimumab 3 mg/kg and nivolumab 1 mg/kg Q3W	Arm A: 3/12 (25%) Arm B: 5/11 (45%)	N/A	Arm A: 25% Arm B: 73%	Arm A: 8% Arm B: 73%	Arm A: 1.5 yr 58% Arm B: 1.5 yr 82%		Arm A: 1.5 yr 67% Arm B: 1.5 yr: 91%	Arm A: 2 yr 76% Arm B: 2 yr 100%
OpACIN-neo, Rozeman et al.	32 months	86	Arm A: max 2 courses of ipilimumab (3 mg/kg) + nivolumab (1 mg/kg) Q3W Arm B: max 2 courses of ipilimumab (1 mg/kg) + nivolumab (3 mg/kg) Q3W Arm C: max 2 courses of ipilimumab (3 mg/kg) Q3W, followed by max 2 courses of nivolumab (3 mg/kg) Q2W	37 (43%)	52 (60%)	64 (74%)	Arm A: 40% Arm B: 20% Arm C: 50%	N/A	N/A	N/A	N/A
OpACIN-neo, Versluis et al.	47 months	86	Arm A: max 2 courses of ipilimumab (3 mg/kg) + nivolumab (1 mg/kg) Q3W Arm B: max 2 courses of ipilimumab (1 mg/kg) + nivolumab (3 mg/kg) Q3W Arm C: max 2 courses of ipilimumab (3 mg/kg) Q3W, followed by max 2 coursed of nivolumab (3 mg/kg) Q2W	37 (43%)	52 (60%)	64 (74%)	Arm A: 40% Arm B: 20% Arm C: 50%	Arm A: 3 yr 87% Arm B: 3 yr 79% Arm C: 3 yr 79% Total 3 yr 82%	Arm A: 3 yr 87% Arm B: 3 yr 77% Arm C: 3 yr 81%	Arm A: 3 yr 87% Arm B: 3 yr 77% Arm C: 3 yr 81% Total 3 yr 88%	Arm A: 3 yr 90% Arm B: 3 yr 93% Arm C: 3 yr 92% Total 3 yr 92%
PRADO, Reijers et al.	28.1 months	99	Max 2 courses of ipilimumab (1 mg/kg) + nivolumab (3 mg/kg) Q3W Personalized Response-Driven Adjuvant Therapy	48 (49%)	60 (61%)	71 (72%)	22%	2 yr 93% MPR 2 yr 64% pPR 2 yr 71% pNR	N/A	2 yr 98% MPR 2 yr 64% pPR 2 yr 76% pNR	N/A
SWOG-1801	14.7 months	313	Arm A: Max 18 courses of adjuvant pembrolizumab 200 mg Q3W Arm B: max 3 courses of neo-adjuvant pembrolizumab 200 mg Q3W and max 15 courses of adjuvant pembrolizumab 200 mg Q3W after surgery	40 (38%)	56 (53%)	82 (79%)	Arm A: 14% Arm B: 12%	N/A	Arm A: 2 yr 49% Arm B: 2 yr 72%	N/A	N/A
Amaria et al.	24.4 months	30	Max 2 courses of neo-adjuvant combination nivolumab 480 mg + relatlimab 160 mg Q4W, followed by max 10 courses of adjuvant nivolumab 480 mg Q4W	17 (56.7%)	19 (63%)	21 (70%)	0%	2 yr 82%	2 yr 81%	N/A	2 yr 88%

FUP: Follow-up; pCR: pathologic complete response (=% viable tumor cells); MPR: major pathologic response (=pCR + near-pCR; ≤10% viable tumor cells); ORR: Overall Response Rate (=pCR + near-pCR + pPR ≤ 50% viable tumor cells); pPR: pathologic partial response (10–50% viable tumor cells); pNR: pathologic non-response (>50% viable tumor cells); TRAE: Treatment-Related Adverse Event; RFS: recurrence-free survival; EFS: event-free survival; DMFS: Distant Metastasis-Free Survival; OS: Overall Survival; N/A: Not Available.

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
