# Peer review of "Neo-Adjuvant Therapy for Metastatic Melanoma"

_cancers, 2024, doi:10.3390/cancers16071247_

Round 1

Reviewer 1 Report (Previous Reviewer 4)

Comments and Suggestions for Authors

The authors included all my suggestions.

Reviewer 2 Report (Previous Reviewer 2)

Comments and Suggestions for Authors

Now it is ready for publication. 

This manuscript is a resubmission of an earlier submission. The following is a list of the peer review reports and author responses from that submission.

Round 1

Reviewer 1 Report

Comments and Suggestions for Authors

Great job.

Please correct in line 142 "response"

Reviewer 2 Report

Comments and Suggestions for Authors

In the present manuscript try to describe recent approaches of new adjuvant therapy. Overall manuscript quite ok but several  graphical illustration  required for broader prospect as per reader view. Author need to mention the very recent clinical data. Also added few of future direction with other type of cancer &  proper justification. 

Reviewer 3 Report

Comments and Suggestions for Authors

It is a short and comprehensive review which makes the article easy to read.

Reviewer 4 Report

Comments and Suggestions for Authors

In this manuscript, the authors provide a review about neo-adjuvant systemic (NAS) therapy in high risk stage III and resectable stage IV melanoma for immunotherapy (single and combinatory) and BRAF/MEKi therapy. This is interesting and relevant especially for traslational researchers and clinicians. The manuscript is well and timely written.

A minor commetn: Section 3 could be improved by including a summary table of all the clinical trial discussed, and when correspond, the comparison among the results from neo-adjuvant vs others (adjuvant) or single treatment vs combinations.

Also, change IFN-y for gamma.